# Full-Scale Train-to-Train Impact Test and Multi-Body Dynamic Simulation Analysis

**Hui Zhao [1], Ping Xu [1], Benhuai Li [1,2], Shuguang Yao [1], Chengxing Yang [1,*], Wei Guo [1] and Xianliang Xiao [1]**

1     Key Laboratory for Track Traffic Safety of Ministry of Education, Central South University, Changsha 410075, China; 184201008@csu.edu.cn (H.Z.); xuping@csu.edu.cn (P.X.); libenhuai@cccar.com.cn (B.L.); ysgxzx@csu.edu.cn (S.Y.); 194212077@csu.edu.cn (W.G.); xiaoxianliang@csu.edu.cn (X.X.)

2     CRRC Changchun Railway Vehicles Co., Ltd., Changchun 130062, China

\*     Correspondence: Chengxing_Yang_Hn@163.com

**Abstract:** When a train crashes with another train at a high speed, it will lead to significant financial losses and societal costs. Carrying out a train-to-train crash test is of great significance to reproducing the collision response and assessing the safety performance of trains. To ensure the testability and safety of the train collision test, it is necessary to analyze and predict the dynamic behavior of the train in the whole test process before the test. This paper presents a study of the dynamic response of the train in each test stage during the train-to-train crash test under different conditions. In this study, a 1D/3D co-simulation dynamics model of the train under various load conditions of driving, collision and braking has been established based on the MotionView dynamic simulation software. The accuracy of the numerical model is verified by comparing with a five-vehicle formations train-to-train crash test data. Sensitivities of several key influencing parameters, such as the train formation, impact velocity and the vehicle mass, are reported in detail as well. The results show that the increase in the impact velocity has an increasing effect on the movement displacement of the vehicle in each process. However, increasing the vehicle mass and train formation has almost no effect on the running displacement of the braking process of the traction train. By sorting the variables in descending order of sensitivity, it can be obtained that impact speed > train formation > vehicle mass. The polynomial response surface method (PRSM) is used to construct the fitting relationship between the parameters and the responses.

**Keywords:** train-to-train crash test; multi-body dynamic; parametric analysis; response surface method

## 1. Introduction

Railway collision accidents have received increasing attention in recent years [1–3]. The collision between trains is an extremely complex and nonlinear process, often accompanied by instability forms such as climbing, collision–induced derailment and overturning after derailment [4]. Carrying out a full-scale train-to-train crash test is the most effective and convincing method, which is of great significance for scientifically reproducing the collision response of trains and truly assessing the safety performance of train collisions [5,6]. However, the research on train crash test is not only extremely expensive but also has low repeatability and certain safety risk. To ensure the testability and safety of the train collision test, the collisions process between vehicles must occur at designated locations. Therefore, it is necessary to perform accurate simulation calculations for the entire process of train driving process, collision process and braking process before the test to determine the location of the collision under different test scenarios and working conditions.

To realize the accurate simulation and prediction of the movement of the entire train collision test, it is necessary to build a whole-process simulation calculation model to simulate the movement and deformation of the train during acceleration, collision and braking. Multi-body dynamic (MBD) method has strong advantages in train system

kinematics and dynamics analysis, which normally provide satisfying time efficiency and tolerable accuracy in calculation. It has been widely adopted in multi-vehicle train dynamic simulations [7,8].

Train traction and dynamic braking have evolved over many years. At present, the longitudinal dynamics model of trains has been widely used in the prediction of vehicle speed and position, study of train configuration and driving cycles and the design of braking systems [9,10]. Wu et al. summarized the current modelling methods used in the study of longitudinal dynamics of trains, including the look-up table methods, equation-based methods and co-simulation methods [11]. Considering the constant adhesion limit, Martin and Hay developed a mathematical model that approximates the traction/dynamic braking force with the ratio of the power of the locomotive to the velocity of the locomotive [12]. An advanced model of traction with the use of the relation between locomotive drawbar forces and velocity for various throttle and dynamics settings has been used by Cole in [13]. Yang [14] establish an 1D and 3D coupled train dynamic model, in which the dynamic behaviours of draft gears and brake shoes are both considered. This study provides a feasible approach to solving the rapidly increasing freedoms considered in train system dynamics.

The multi-body dynamics method has also been widely used in multi-vehicle train collision simulation for collision energy management and train collision posture response research. K. Severson established an MBD model that simulates the three-dimensional rigid body motion of the vehicle [15]. The longitudinal motion simulation results obtained are basically consistent with the full-scale test results, but the vertical and lateral motions are quite different from the test results. J.P. Dias and M.S. Pereira presented a design methodology for crashworthy structures based on multi-body dynamics model. The design methodology is applied to single and multi-criteria optimization of train sets in different collision scenarios [16–18]. Ambrósio et al. take the force-displacement curve of the buffers, couplers, structural devices and suspension elements based on simulation and test as the input of MBD model, and then compare and correct the collision MBD model [19]. Sun et al. study the relationship between the length and impedance force parameters of crush components in HE-zone and the maximum deceleration of vehicle centre of gravity based on the MBD model of rail vehicles [20]. Zhou et al. study the influence of parameters (e.g., collision mass, nodding frequency and the height of the centre of gravity from the rail surface) on the overriding and derailment of rail vehicles, and give the measures to reduce the risk of them [21–23]. Yang et al. developed a three-dimensional MBD model of crashed vehicles coupled with moving tracks to research the dynamic behaviour of the train front-end collision on tangent tracks [24]. Ling et al. investigated the derailment responses of trains under frontal collision on road trucks obliquely stuck on rail tracks at level crossings, sensitivities of several key influencing parameters [25,26].

It can be seen that the one-dimensional dynamics calculation model has high efficiency in the calculation of the train driving and braking process, while the three-dimensional dynamics calculation model can simulate the complex instability movement during the train collision. However, the train-to-train collision test is a complex dynamic problem, including multiple load conditions, such as vehicle traction, collision and braking. The simulation boundary conditions and calculation time steps of each working condition are different. It is difficult to calculate the whole process of train-to-train collision process with the same dynamic model.

Therefore, this paper proposes a method of 1D/3D co-simulation to simulate and analyse the whole process of train-to-train collision process. High-precision 3D dynamic model can meet the requirements of complex train collision dynamics analysis, and the 1D dynamic model of train driving and braking conditions, which can reduce the amount of three-dimensional calculations and save calculation costs.

In this paper, a 1D/3D co-simulation dynamic model for the entire process of train-to-train collision process is constructed in Section 2. In Section 3, a full-scaled train-to-train crash test is conducted and the accuracy of the simulation model is verified by compared

with the test results. Section 4 shows the effects of the train formation, vehicle mass and the collision velocity on the dynamic response of the train at each stage. Sensitivities of several key factors are presented in detail as well. Section 5 uses the polynomial response surface method to construct the fitting relationship between the parameters and the responses.

## 2. Simulation

Numerical model for the entire process of train-to-train impact is formulated using the 1D/3D coupled dynamic simulation method, and the dynamic behaviour of the train under traction, braking, and collision load conditions is studied.

### 2.1. Vehicle Movement Process

The procedure of the train-to-train crash process is complicated and can be divided into five phases. The flow chart of the train-to-train crash process is shown in Figure 1.

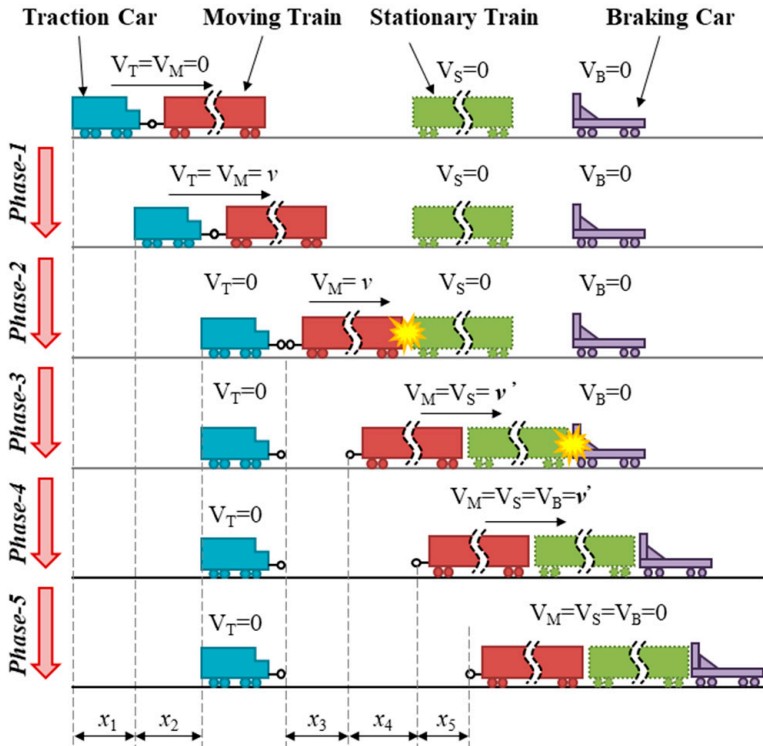

**Figure 1.** Schematic diagram of train multi–group collision process.

- Phase 1: The traction car driving the moving test train accelerates to the target velocity of the crash test;
- Phase 2: The traction car releases the moving test train. While the moving test train continues to slide forward, the locomotive takes braking measures to protect the safety of the driver;
- Phase 3: Under the protection of multiple safety protection systems, the moving test train collides with a row of the identical stationary train. After the collision, part of the energy is absorbed by the plastic deformation of the energy-absorbing structure, but part of the energy is still stored in the form of kinetic energy, which makes the moving train and the stationary train continue to slide forward;
- Phase 4: After the train-to-train crash test, the sliding test train collides with the stationary braking car. After the collision, the braking car will slide forward together with the test train;
- Phase 5: Under the action of the braking force of the braking car, the test train and the braking car decelerate together until it stops.

The objective of this research is to formulate appropriate test plans according to vehicle motion in each process of the test. Therefore, the maximum displacement response of the vehicle in each process is taken as the target response, which is defined as $x_1$, $x_2$, $x_3$, $x_4$ and $x_5$, respectively.

### 2.2. 1D Dynamic Model of Train System

In the load condition of train traction and braking, since the entire process of train movement occurs on a straight line and remains on the line, the impact of the vertical and lateral movement of the vehicle in this process can be ignored. Therefore, it is appropriate to use the longitudinal multi-body dynamics model when calculating train traction and braking conditions.

When analysing the longitudinal dynamic behaviour of the train, the train is usually abstracted as a mass spring damping system with multiple mass points on a one-dimensional linear coordinate. Each vehicle is simplified into a mass point model, and the mass point is located at the midpoint of the length of the train. For a multi-vehicle system, the longitudinal dynamics equation for the *i*th vehicle is expressed as:

$$M_i^{1d}\ddot{q}_i^{1d} = Q_i^{1d} \tag{1}$$

where $M_i^{1d}$, $\ddot{q}_i^{1d}$ and $Q_i^{1d}$ are the mass column matrix, the acceleration column matrix, and the column matrix of external forces of the one-dimensional vehicle subsystem acting on the vehicle subsystem. External forces include traction force ($F_T$), braking force ($F_B$), and coupler force between adjacent vehicles ($F_{cp}$).

The force of coupler buffer device between the car bodies is simulated by a spring-damper unit with nonlinear hysteresis characteristics connecting adjacent mass points, the loading and unloading curves of the head- and middle-car buffers are shown in Figure 2.

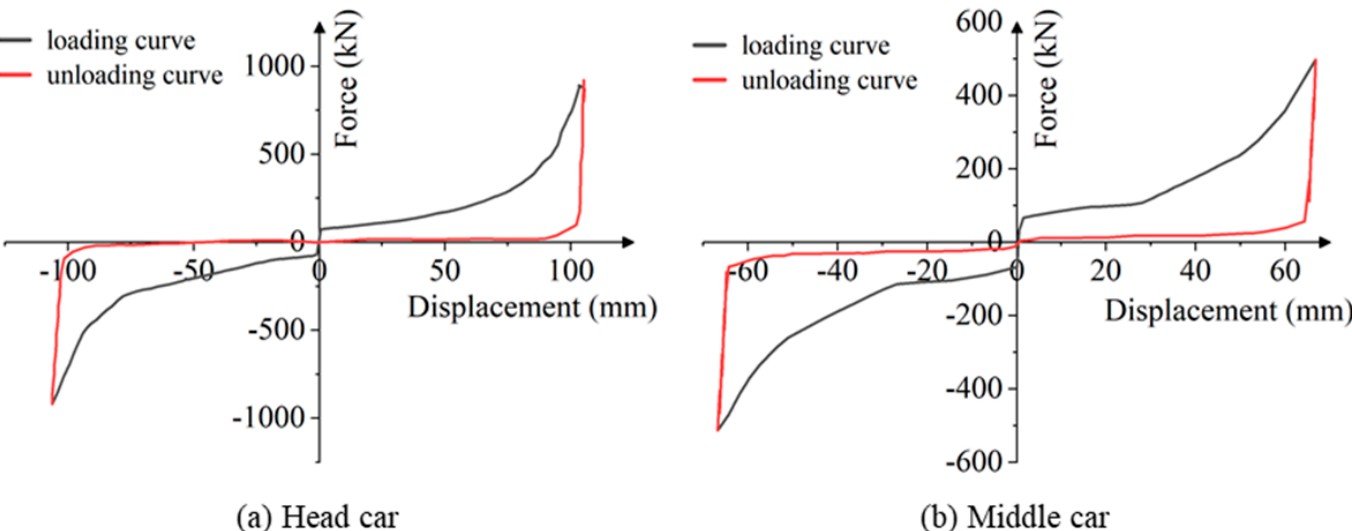

**Figure 2.** Loading and unloading curves of buffer.

The traction and braking force acting on the vehicle are all applied to the mass point in the form of force elements. The traction force is determined based on manufacturers' locomotive performance curves, as shown in Figure 3.

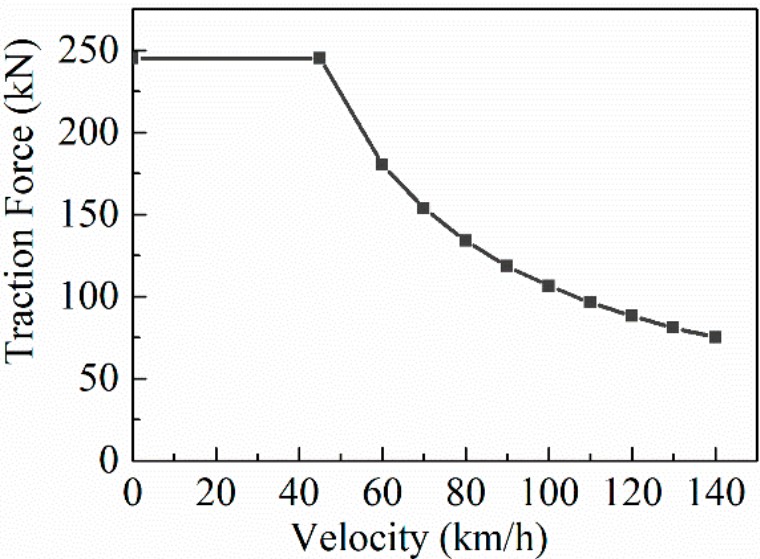

**Figure 3.** The relationship between the traction force and the vehicle speed.

The braking process of the train is simplified into an idling process and an effective braking process [27]. The braking idling process is the process of the train sliding at the initial braking velocity in the state of no braking force, and the braking time is taken as 2.5 s; the effective braking process is the process of the train decelerating under the action of braking force, and the braking force calculation of this process adopts conversion. The braking force calculation adopts the conversion brake–shoe pressure calculation method, as shown in formula:

$$F_B = K_h \varphi_h \tag{2}$$

where $K_h$ represents the converted brake–shoe pressure, taken as 650 kN; $\varphi_h$ represents the friction factor between the brake–shoe and the vehicle, which can be expressed as:

$$\varphi_h = 0.372 \times \frac{17v + 100}{60v + 100} + 0.012(120 - v') \tag{3}$$

where $v$ represents the current running velocity (km/h), $v'$ represents the initial braking velocity (km/h)

According to the simplification above, the 1D train dynamic model is established. The advantage of this model is that no matter which car of the train is considered, the coupler force and vehicle state that change with the simulation time can be easily and quickly obtained.

### 2.3. 3D Dynamic Model of Train System

During a train collision, there is not only a longitudinal impact load between the train and the vehicle, but also lateral, vertical load and torque in three directions, causing the train to derail due to head shaking, rolling, and rolling, and nodding, floating and sinking. Climbing, collapse, and other system instability is caused by expansion and contraction. Therefore, when calculating and simulating train collision conditions, it is necessary to consider the laws of train longitudinal, vertical and lateral movement to construct a three-dimensional train dynamic analysis model.

According to the theory of vehicle dynamics [28], the calculation model of a high-speed vehicle is shown in Figure 4. In the coupled dynamic model, each vehicle is modelled as a 42 degrees of freedom (DOF) nonlinear multi-body system, which includes seven rigid components: a car body, two bogies, and four wheelsets. Each component of the vehicle has 6 DOFs: the longitudinal displacement $x$, the lateral displacement $y$, the vertical displacement $z$, the roll angle $\varphi$, the pitch angle $\beta$, and the yaw angle $\psi$.

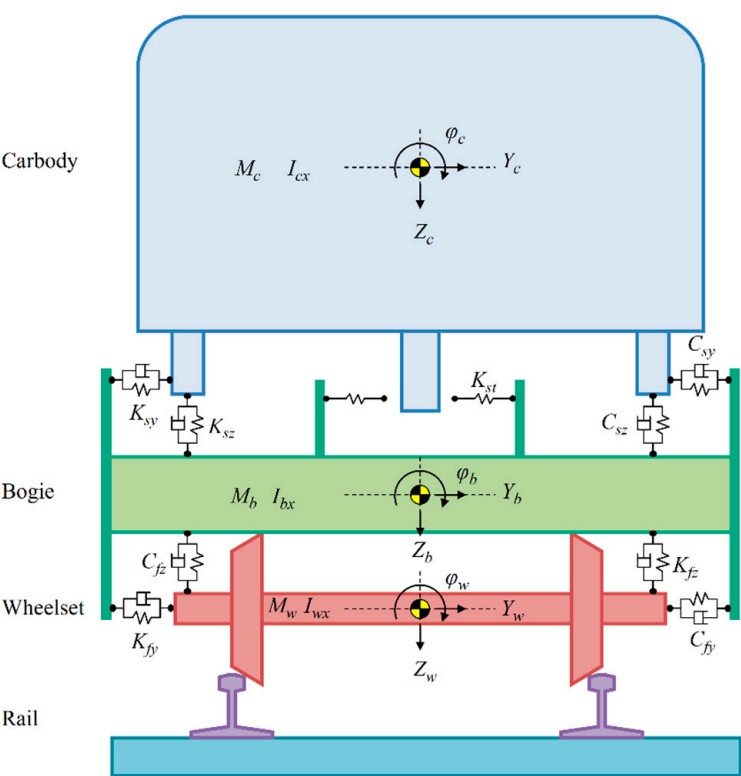

**Figure 4.** Vehicle system dynamics model based on multi–rigid body theory.

In Figure 4, axis x is in the moving direction of the train, axis z is in the vertical direction, and axis y is in the lateral direction of the track. The notations *C* and *K* with subscripts stand for the coefficients of the equivalent dampers and the stiffness coefficients of the equivalent springs, respectively.

The motion equations of the *k*–th vehicle can be expressed as [25]:

$$M_k \ddot{X}_k(t) + C_k(\dot{x}_k)\dot{X}_k(t) + K_k(\dot{x}_k)X_k(t) = F_{wrk}(\dot{x}_k, x_k, \dot{x}_t, x_t, v, t) + F_{ink}(\dot{x}_k, x_k, t) \quad (4)$$

where $M_k$ is the mass matrix of the vehicle *k*, $C_k$ and $C_k$ are the damping and stiffness matrices depending on the current state of the vehicle subsystem to describe nonlinearities within the suspension; $X_k(t)$, $\dot{X}_k(t)$, and $\ddot{X}_k(t)$ are the displacement, velocity, and acceleration vectors, respectively; $\dot{x}_k, x_k, \dot{x}_t, x_t$ are the displacement and velocity vectors related to the suspension forces and wheel/rail contact forces, respectively; *v* is the train velocity; $F_{wrk}$ is the vector of the nonlinear wheel–rail contact forces, which is determined by the displacements and velocities of the vehicle *k* and the track. $F_{ink}$ is the vector of the nonlinear inter-vehicle contact forces due to the couplers and anti-climbers.

In order to simulate the whole process of train-to-train impact test, Motionview dynamics software (Version 2021.1) is used to establish a 3D dynamics simulation model of the vehicle. The vehicle dynamic model consists of five subsystems: the vehicle subsystem, the energy-absorbing subsystem and the wheel/rail contact subsystem.

### 2.3.1. Modelling Vehicle Subsystem

The basic parameters of the test vehicle are modelled with reference to a high-speed train in China. The wheelsets and the bogies are connected by the primary suspensions, while the car body is supported by the bogies through the secondary suspensions. Nonlinear 3D spring–damper elements were used to build the suspension model [29]. Figure 5 shows the vehicle model of the head car and the middle car of the high-speed train. The basic parameters of the vehicle are shown in the Table 1.

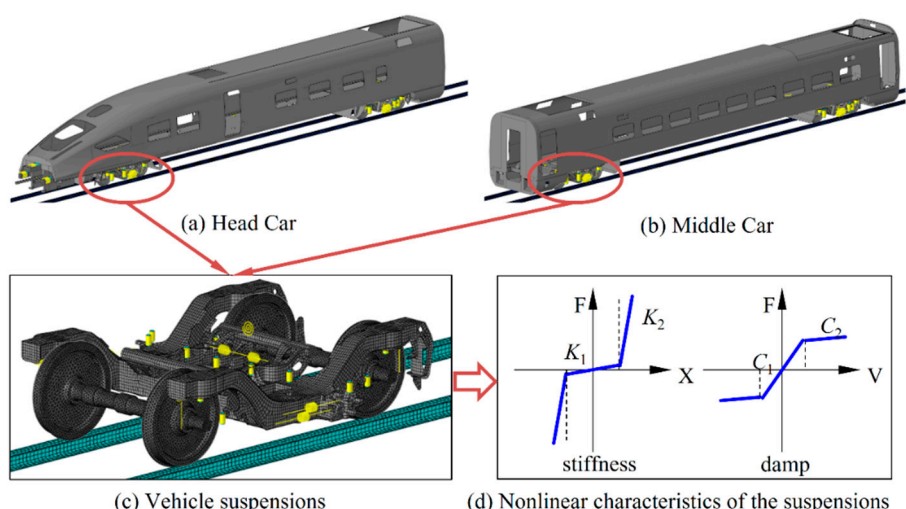

**Figure 5.** A vehicle dynamics model constructed with reference to a high–speed train.

**Table 1.** Basic parameters of the high-speed train.

| Type | Parameters | Unit | Value |
|---|---|---|---|
| Geometric parameters | Vehicle distance | mm | 17,800 |
| | Bogie axis distance | mm | 2500 |
| | Rail distance | mm | 1435 |
| | Wheel rolling circle diameter | mm | 860 |
| | Wheel profile shape | | LMA |
| | Rail profile shape | | CHN60 |
| Inertia parameter | Carbody mass | t | 43.0 |
| | Carbody moment of inertia-X | kg·m$^2$ | 69,171 |
| | Carbody moment of inertia-Y | kg·m$^2$ | 1,347,125 |
| | Carbody moment of inertia-Z | kg·m$^2$ | 1,349,239 |
| | Bogie mass | t | 3.660 |
| | Bogie moment of inertia-X | kg·m$^2$ | 3876 |
| | Bogie moment of inertia-Y | kg·m$^2$ | 8098 |
| | Bogie moment of inertia-Z | kg·m$^2$ | 10,248 |
| | Wheelset mass | t | 2.816 |
| | Wheelset moment of inertia-X | kg·m$^2$ | 2024 |
| | Wheelset moment of inertia-Y | kg·m$^2$ | 452 |
| | Carbody moment of inertia-Z | kg·m$^2$ | 2024 |
| Suspension parameters | Axlebox spring longitudinal stiffness | N/mm | 886 |
| | Axlebox spring lateral stiffness | N/mm | 919 |
| | Axlebox spring vertical stiffness | N/mm | 919 |
| | Primary suspension vertical damper | N·s/mm | 20 |
| | Airspring longitudinal stiffness | N/mm | 107 |
| | Airspring lateral stiffness | N/mm | 107 |
| | Airspring vertical stiffness | N/mm | 173 |
| | Secondary suspension longitudinal damper | N·s/mm | 810 |
| | Secondary suspension lateral damper | N·s/mm | 15 |
| | Secondary suspension vertical damper | N·s/mm | 10 |

### 2.3.2. Modelling Energy-Absorbing Subsystem

The energy-absorbing structure between vehicles are respectively equivalent to nonlinear springs, and their mechanical characteristics are described by defining the simplified load–displacement characteristic curve [30,31], shown as in Figure 6.

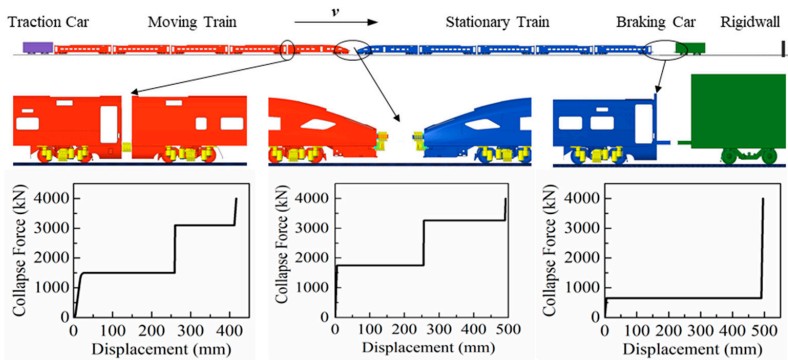

**Figure 6.** Energy-absorbing and simplified load–displacement curve.

### 2.3.3. Modelling Wheel/Rail Contact Aubsystem

In the 3D dynamic model constructed in this paper, the interaction between the wheelset and the rail is simulated by a general "Mesh–to–Mesh Contact Detection Method". The current wheel-rail contact model is set up based on 60 kg/m rails and the LMA profiles [32,33]. The wheel/rail normal contact force is calculated based on a nonlinear Hertz contact model, and the tangential contact force of the wheel and rail is simulated by the Coulomb friction model.

### 2.4. Simulation Result

The dynamic response of each vehicle changes with time during the whole process of the train-to-train crash is shown in the Figure 7. In phase 1, the traction car promotes 5 vehicles to accelerate, and the running distance and running velocity of the train increase with time. The duration time of this phase is about 20.6 s, and the movement distance is about 85.8 m. In phase 2, when the traction car brakes alone, the first step is the brake idling process. At this same time, the running velocity of each vehicle remains basically unchanged, and the running displacement increases linearly. Then the effective braking process is carried out, the running velocity of the traction car is gradually reduced, and the velocity of the test vehicle remains unchanged. The duration time of this process is about 76 s, and the movement distance is about 43.8 m. In phase 3, the duration time of the collision process between the moving train and stationary train is about 0.6 s. The velocity of the moving train gradually decreases, and the velocity of the stationary train gradually increases. Until the velocity of the moving train and the stationary train are basically the same, and the collision process ends. The movement distance of this process is about 4.1 m. In phase 4, the moving train and the stationary train collided with the braking car. After about 0.47 s, the two trains moved at the same velocity and began to slide in the same direction. The movement distance in this process is about 1.9 m. In phase 5, all vehicles brake together. After about 10.3 s, all vehicles are braked. The movement distance of this process is 25.6 m.

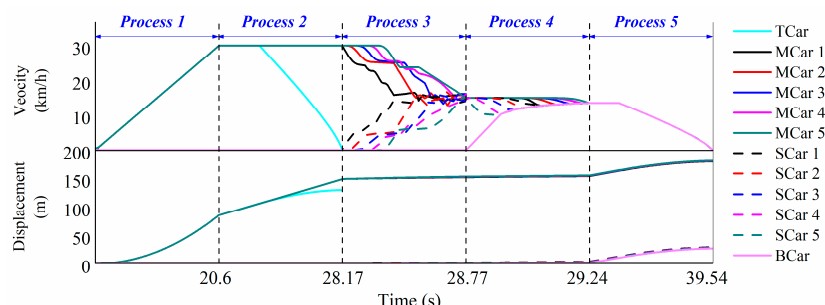

**Figure 7.** Dynamic response of each vehicle during the whole process of vehicle collision based on dynamic simulation result.

## 3. Test

### 3.1. Test Setup

A train-to-train crash test is performed at the train-to-train collision test bench developed by CRRC Changchun Railway Vehicles Co., Ltd., Changchun, China as indicated in Figure 1. According to the requirements of the EN15227 [34] standard, the collision scene of the test is set as a front-end impact between two identical trains. In this test, a traction car driving five-vehicle formation Chinese high-speed train, traveling at a velocity of 30 km/h, collided with a stationary identical five-vehicle formation high-speed train on a straight track. The definition of moving/stationary vehicle number and collision interface number is also shown in Figure 8. Table 2 shows the vehicle mass of traction car, moving/stationary train and braking car.

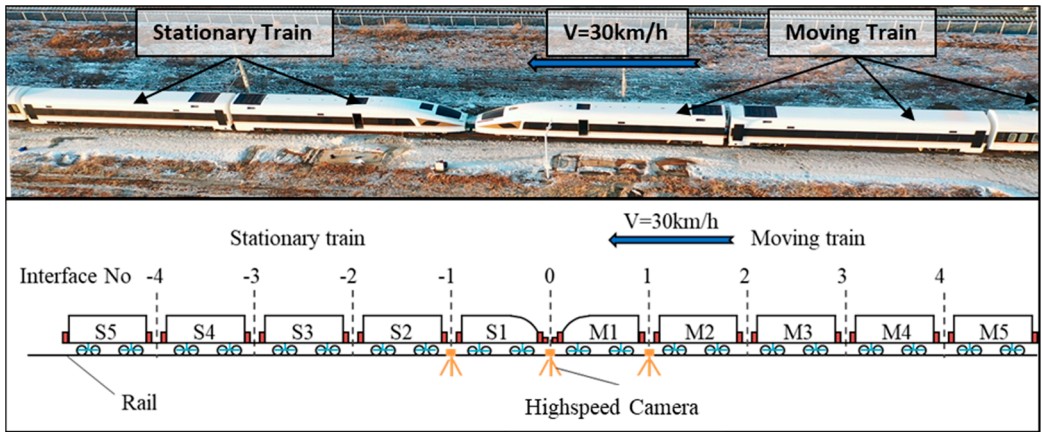

**Figure 8.** Experimental crash condition.

**Table 2.** Train mass distribution in the test.

| Train Type | Traction Car Mass | Moving/Stationary Train Mass | | | | | Braking Car Mass |
|:---:|:---:|:---:|:---:|:---:|:---:|:---:|:---:|
| | | 1 | 2 | 3 | 4 | 5 | |
| Mass (t) | 150 | 52.9 | 60.6 | 60.4 | 50 | 50 | 60 |

The vehicle-mounted GPS velocimeter installed on the car body can measure the velocity and displacement of the test vehicle during the whole process. Three high-speed digital cameras are arranged on the collision interface and the two adjacent interfaces to capture the impact process. The time histories of the displacement and the velocity can be obtained from the high-speed digital camera.

### 3.2. Test Result

#### 3.2.1. Vehicle Movement in Each Process

- Phase 1: In the process of the traction car pushing the five moving test train and accelerating to 30 km/h, the movement displacement of the driving car is about 90 m. During this test, the acceleration process of the driving car was manually controlled by the driver. The average traction power of the locomotive starting is 70% of the design maximum traction power of the locomotive. In future research and tests, the locomotive will start with full power.
- Phase 2: The braking process of the traction car first experienced a braking idling time of 2.5 s, and the braking idling distance is about 21 m; then it was decelerated to 0 km/h with a braking displacement of 23 m. The total braking distance of the driving car is 44 m.
- Phase 3: The duration time of the impact process between the moving train and stationary train is about 0.6 s. During this process, the maximum running displacement

of the moving train is 4 m. After the collision, the moving train and the stationary train glide forward together at a velocity of about 15 km/h.

- Phase 4: The moving train and the stationary train collided with the braking car, and the maximum running displacement is 2 m. After the collision, the moving train, the stationary train and the braking train glide forward together at a velocity of about 13.5 km/h.
- Phase 5: The braking process of the moving train, the stationary train and the braking train first experienced a braking idling time of 2.5 s, and the braking idling distance is 9 m; then it was decelerated to 0 km/h, and a braking displacement of 17 m is required for effective braking. The total braking distance of the driving car is 26 m.

Figure 9 shows the ratio of the vehicle displacement in each phase to the total displacement during the crash test. The running displacements of Phase 1, 2 and 5 account for a large proportion of 54.2%, 26.5% and 15.7%, respectively, which have a greater influence on the total running displacement. However, the sum of the proportions of phase 3 and 4 is less than 5%, and the contribution to the total running displacement is small. This is because compared with the driving and braking process, the impact process is an instantaneous process, and duration is extremely short, resulting in smaller movement displacement.

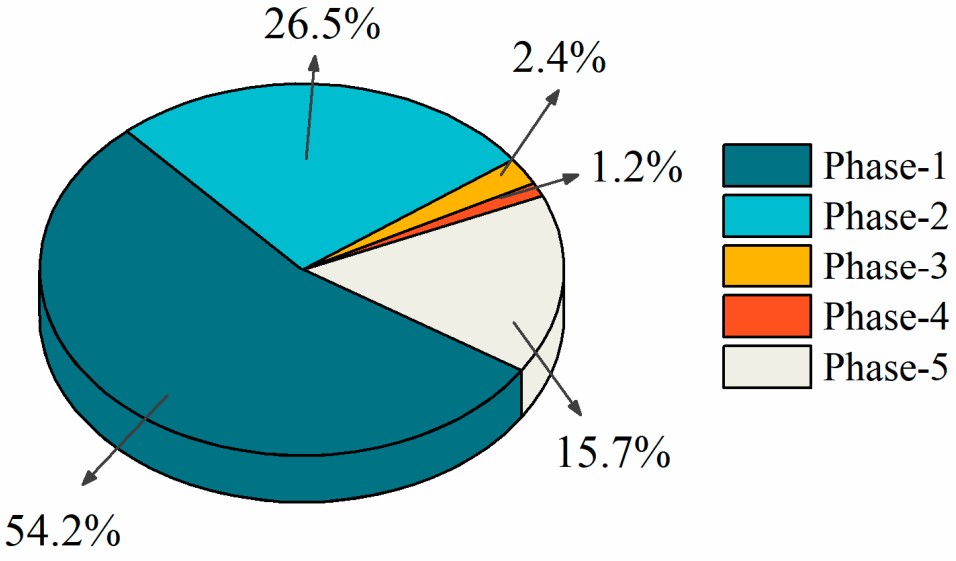

**Figure 9.** Displacement distribution in each process of crash test.

### 3.2.2. Collision Analysis

During the train-to-train crash test, the vehicle structures of test train remained essentially intact. There is no loss of occupant volume for the passengers and there is no override between the adjacent cars. Nearly all the damage was focused on the energy-absorbing structures in a controlled manner.

Figure 10 shows a series of photographs taken from a high–speed movie of the test. When the two trains collide at the speed of 30 km/h, the front end of the two trains firstly starts to contact. While the primary energy-absorbing structure and secondary energy-absorbing structure of the two trains are triggered step by step to dissipate kinetic energy. As the speed of M1 car decreases, the coupler buffer of interface 1 starts to work, and at the same time, as the speed of S1 car increases, the coupler buffer of interface -1 starts to work. The coupler buffer and anti-climbing device between the front car and the rear car acts in turn, and so on. At the end of the 0.6 s collision, the speed of the two trains tends to be the same. Currently, the vehicle still has a residual speed, which is about 15 km/h.

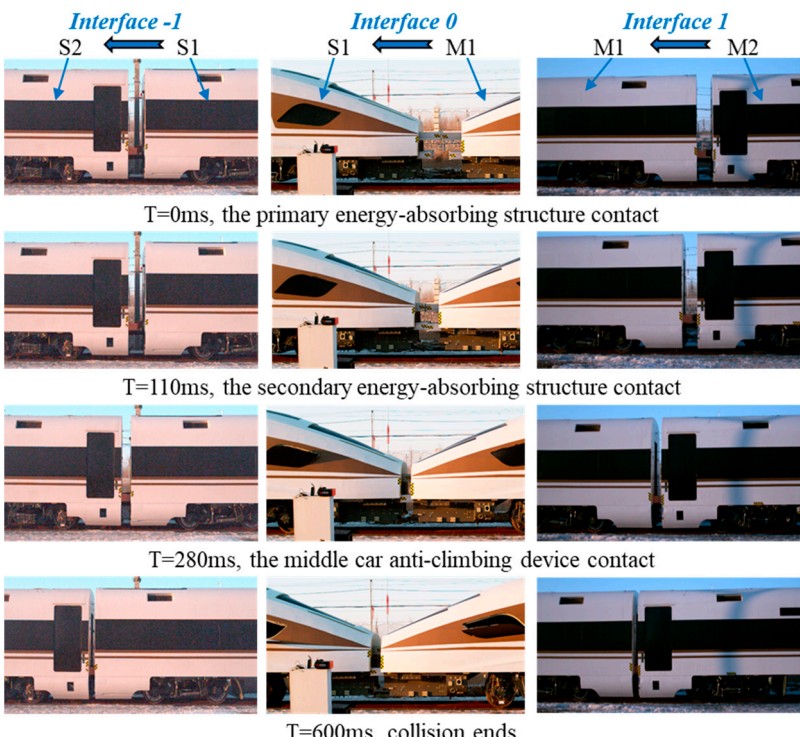

**Figure 10.** Sequential photos of the in–line train-to-train crash test.

### 3.3. Validation of the Simulation Model

To verify the accuracy of the simulation model, the collision conditions of two trains with five-grouping are selected for simulation calculation and compared with the test results of the full-scale test.

#### 3.3.1. Vehicle Movement in Each Process

Table 3 shows the vehicle movement distance comparison of each process between the dynamic simulation results and the experimental measurement values. We can from that the movement distance error of each process does not exceed 5% at most. Therefore, it can be proved that the 3D train dynamics model constructed in this paper can accurately predict the movement of the vehicle during the whole test.

**Table 3.** Displacement comparison of each process of experiment and simulation.

| Response | $x_1$ | $x_2$ | $x_3$ | $x_4$ | $x_5$ | $x$–Total |
|---|---|---|---|---|---|---|
| Simulation (m) | 85.8 | 43.8 | 4.1 | 1.9 | 25.6 | 161.2 |
| Test (m) | 90 | 45 | 4 | 2 | 25 | 166 |
| Error (%) | 4.67 | 2.67 | 2.5 | 5 | 2.4 | 2.89 |

#### 3.3.2. Collision Analysis

Analysing the high-speed photography pictures of the collision interface and the two adjacent interfaces, the velocity curve and displacement curve of each car in the shooting range is obtained. Figure 11 shows the animations of the running velocity curve, displacement curve and deformation sequence of each vehicle during the collision. Comparing the data obtained from the dynamic simulation with the data obtained from the test, it can be found that the simulation and the test have the same sequence of actions, and the curve consistency is relatively high.

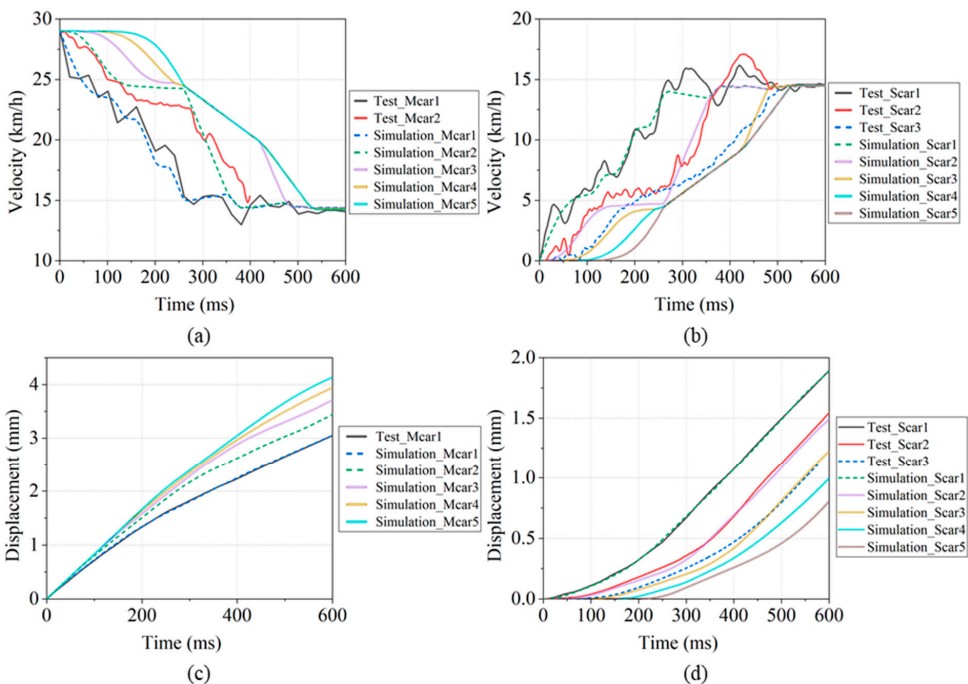

**Figure 11.** The movement of each vehicle during the collision. (**a**) Moving train velocity; (**b**) Stationary train velocity; (**c**) Moving train displacement; (**d**) Stationary train displacement.

The comparative analysis of the above simulation and experiment proves the correctness of the dynamic model constructed in this paper. Therefore, based on the verification, the simulation model can be applied to the next parametric analysis.

## 4. Parametric Analysis

The numerical model constructed in Section 2 is used for dynamic simulation of the whole process of train-to-train crash. Based on the results of 3D train dynamic simulation model verified by experiments, this paper will study the effects of the train formation ($n$), vehicle mass ($m$) and collision velocity ($v$) on the moving response of the train at each stage of collision. The responses $x_1$, $x_2$, $x_3$, $x_4$ and $x_5$ respectively represent the maximum movement displacement of the five process vehicles. The responses $v_1$, $v_3$, $v_4$ and $v_5$ respectively represent the instantaneous velocity of the moving train at the end of the Phase 1, 3, 4, 5.

### 4.1. Effect of Train Formation

Figure 12 illustrates the influence on the displacement response of each process of the train crash process when the train formation is increased from 1 to 8 with a step of 1, in which the collision velocity with 30 km/h and the vehicle mass with 45 t.

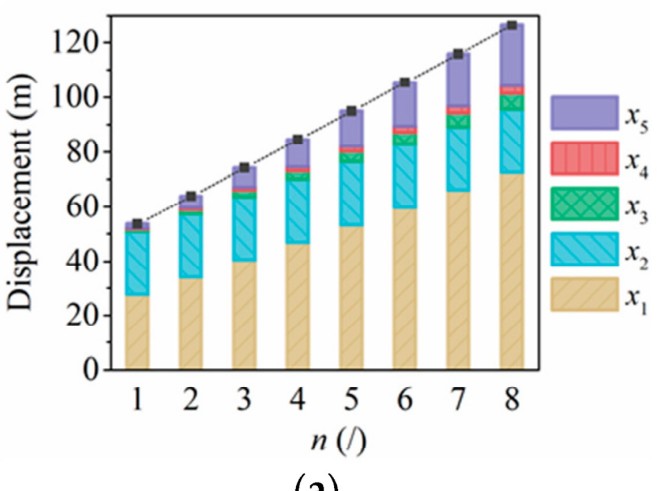 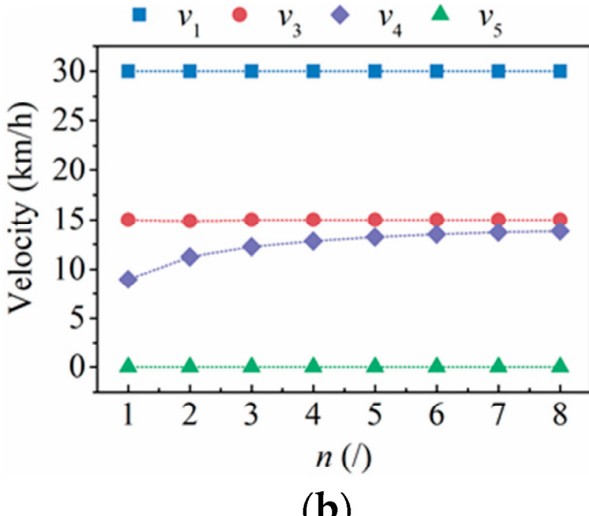

(a)           (b)

**Figure 12.** The influence of the train formation on the dynamic response: (**a**) displacement, (**b**) velocity.

- $x_1$: The displacement $x_1$ increases linearly with the increase in the train formation. The displacement $x_1$ was shown to increase from 27.6 m to 72.3 m when the train formation increased from 1 to 8. This is because when the traction force of the locomotive is constant (for the target velocity is less than 45 km/h), the mass of the object affected by the traction force increases linearly with the increase of the train formation, which causes the acceleration of the vehicle to decrease, so the distance $x_1$ will increase accordingly.

- $x_2$: Since the braking distance of the traction car is only related to its own initial state, changing the train formation of the test train that has been separated from the traction car has no effect on the value of $x_2$.

- $x_3$: As the number of test trains participating in the collision process increases, the system initial total kinetic energy of the collision increases linearly, and the maximum vehicle displacement $x_3$ of the process also increases. However, as the number of train formation increases, the instantaneous velocity $v_3$ of the vehicle at the end of the process does not change and is maintained at about 15 km/h (1/2 of the impact velocity).

- $x_4$: As the train formation increases from 1 to 8, the duration time of the collision process between the test train and the braking car has gradually increased, and the velocity $v_4$ after the collision increased significantly, from 9 km/h to 13.85 km/h. The movement displacement $x_4$ of this process also increases.

- $x_5$: For the post-collision braking process, the increase in the train formation not only increases the mass affected by the braking force, but also increases of the initial braking velocity $v_4$, so $x_5$ will gradually increase with the increase of train formation, and the increase velocity is getting faster and faster.

### 4.2. Effect of Vehicle Mass

Figure 13 illustrates the influence on the dynamic response of each process of the train crash process when the vehicle mass is varied from 30 t to 60 t with a step of 5 t, in which the collision speed with 30 km/h and the train formation with 5.

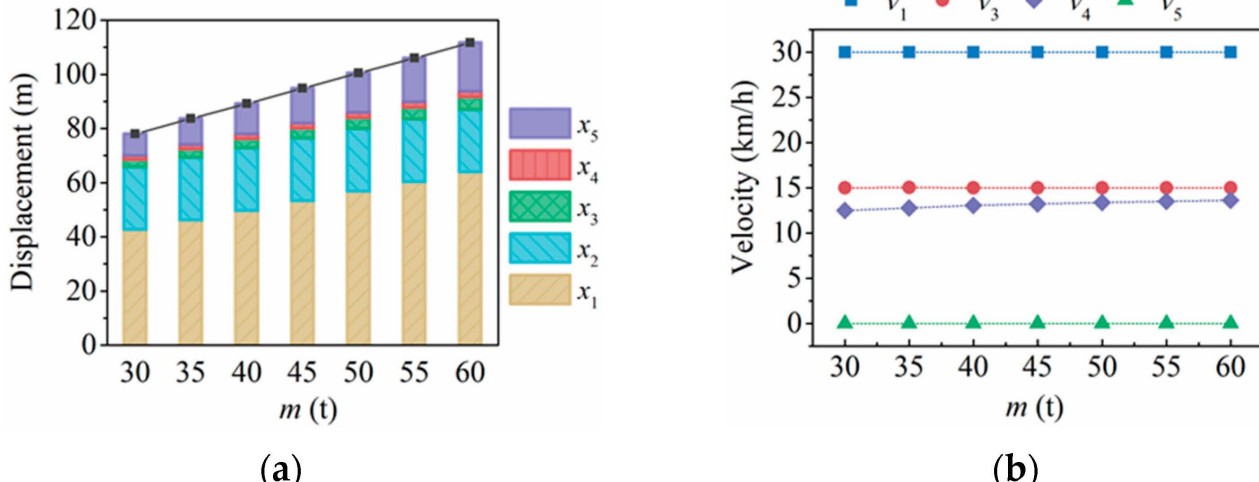

**Figure 13.** The influence of the train mass on the dynamic response: (**a**) displacement, (**b**) velocity.

- $x_1$: The displacement $x_1$ increases linearly with the increase in the vehicle mass. The displacement $x_1$ was shown to increase from 42.52 m to 63.78 m when the vehicle mass increased from 30 t to 60 t. This is because when the traction force of the locomotive is constant (for the target velocity is less than 45 km/h), the mass of the object affected by the traction force increases linearly with the increase of the vehicle mass, which causes the acceleration of the vehicle to decrease, so the distance $x_1$ will increase accordingly.
- $x_2$: Since the braking distance of the traction car is only related to its own initial state, changing the vehicle mass of the test train that has been separated from the traction car has no effect on the value of $x_2$.
- $x_3$: As the train mass participating in the collision process increases, the system initial total kinetic energy of the collision increases linearly, and the maximum vehicle displacement $x_3$ of the process also increases. However, as the vehicle mass increases, the instantaneous velocity $v_3$ of the vehicle at the end of the process does not change, and is maintained at about 15 km/h (1/2 of the impact velocity).
- $x_4$: As the vehicle mass increases from 30 t to 60 t, the duration time of the collision process between the test train and the braking trolley has slightly increased, and the velocity $v_4$ after the collision increased slightly, from 12.48 km/h to 13.63 km/h. The movement displacement $x_4$ of this process also increases.
- $x_5$: According to the above analysis, as the increase of vehicle mass, the initial braking velocity $v_4$ will increase, and at the same time the mass affected by the braking force will increase, so $x_5$ will gradually increase with the increase of vehicle mass, and the increase and the increase velocity is getting faster and faster.

### 4.3. Effect of Impact Velocity

In the simulation, the impact velocity was varied from 5 km/h to 55 km/h, with a step of 5 km/h, the train formation and the vehicle mass were set as 5 and 45 t, respectively. Figure 14 depicts the impact velocity on the dynamic response of each process of the train crash process.

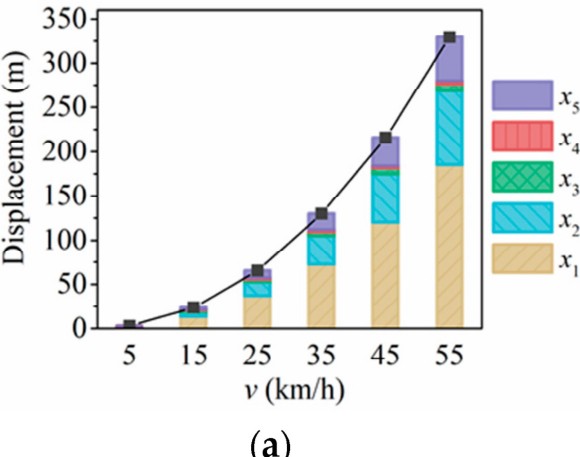
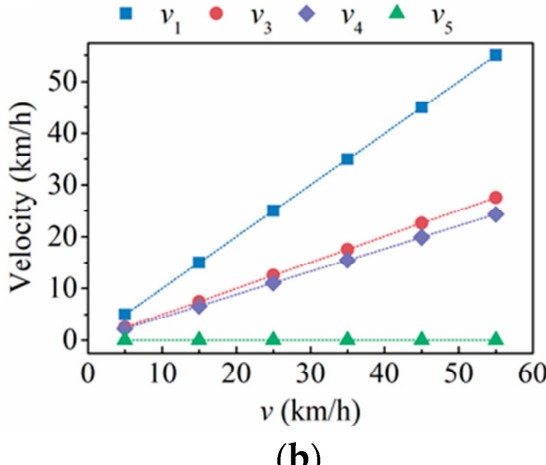

**Figure 14.** The influence of the impact velocity on the dynamic response: (**a**) displacement, (**b**) velocity.

- $x_1$: The displacement $x_1$ increases approximately exponentially as the collision velocity increases. The displacement $x_1$ was shown to increase from 1.48 m to 184.62 m when the collision impact increased from 5 km/h to 55 km/h.
- $x_2$: When the impact velocity increases from 5 km/h to 55 km/h, the braking distance of the traction car increases exponentially from 0.42 m to 84.12 m.
- $x_3$: As the impact velocity increases, the system initial total kinetic energy of the collision increases exponentially, and the maximum vehicle displacement $x_3$ of the process also increases. As the impact velocity increases from 5 km/h to 55 km/h, the displacement $x_3$ increases from 0.31 m to 7.39 m, and the instantaneous velocity $v_3$ of the vehicle at the end of the process also increases, respectively 2.5 km/h, 7.5 km/h, 12.5 km/h, 17.5 km/h, 22.5 km/h and 27.5 km/h.
- $x_4$: As the velocity $v_3$ increases, the duration of the train collision process gradually increases, and the maximum movement displacement $x_4$ of the train also increases dramatically, and the instantaneous velocity $v_4$ of the vehicle at the end of the process increases, respectively 2.2 km/h, 6.6 km/h, 11.0 km/h, 15.4 km/h, 29.9 km/h and 24.3 km/h.
- $x_5$: For the braking process, the change in the initial braking velocity $v_4$ has a great influence on the braking process. $x_5$ will gradually increase with the increase of the impact velocity, and the speed of increase will be faster and faster.

### 4.4. Sensitivity Analysis

Based on the above simulation calculation results, in order to illustrate the influence of different design variables on the vehicle dynamic response, the radar chart of the vehicle displacement response at different stages of train collision is drawn in Figure 15. In order to better compare the influence of each parameter, this study normalized all design variables. The larger the enclosed area in the radar chart, the greater the comprehensive influence on all evaluation design variables.

Sort the design variables according to the sensitivity to displacement response from high to low: impact velocity > train formation > vehicle mass. Increasing vehicle mass and train formation have no effect on the running displacement of the moving vehicle during braking, while the impact velocity change has the greatest influence on the displacement response during the whole process.

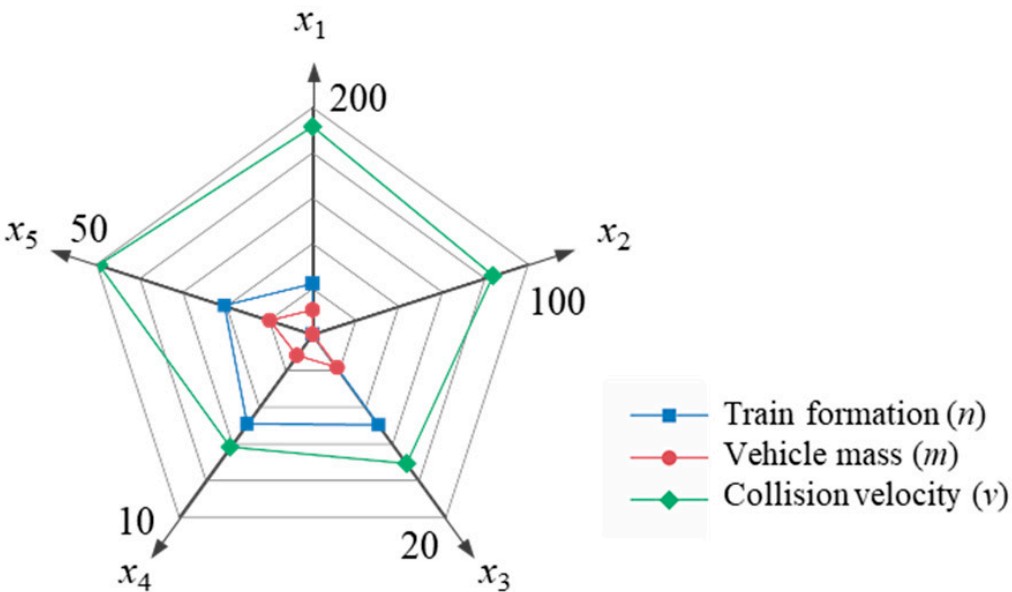

**Figure 15.** Sensitivity of design parameters to the displacement response of each stage.

## 5. Response Surface Models

### 5.1. Design of Experiment

The design of experiments (DOE), the sampling strategy of the surrogate model, determines the number and the spatial distribution of the sampling points. Its goal is to conduct a reasonable statistical analysis of the experimental data based on probability. The hammersley sequence leads to a nearly quadratic reduction in the number of samples compared with the uniform sampling method, and thereby the units of time and cost, while maintaining the same level of accuracy [35]. In this study, the hammersley sampling method was adopted, and 50 levels for each variable were used to fit an accurate surrogate model.

### 5.2. Response Surface Models

Surrogate modelling technique is a kind of efficient measure in approximating design variables and output responses with a moderate number of computational calculations. Currently, widely used surrogate models are response surface (RS), Kriging (KRG) and radial basis function (RBF) models. As an effective alternative, the polynomial response surface (PRS) model has proven to be particularly effective and widely used [36,37].

Considering n–dimensional design matrix $x = [x_1, \dots , x_n]^T$ and objective indicators $y = [y_1, \dots , y_n]^T$, the functional formulation of polynomial RSM (PRSM) can be described by Equation (5).

$$\mathrm{y} = \hat{y} + \varepsilon = \sum_{i=0}^{p-1} a_i \varphi_i(x) + \varepsilon \tag{5}$$

where $\varphi_i(x)$ are the basic function of input variables $x$, $a_i$ are corresponding coefficients of polynomial items, determined by output responses y using the least–square methods, $p$ is the total number of basis function $p = (n + m)!/(n!m!)$, $n$ is the number of design variables and $m$ is the order of PRSM.

To downsize PRSMs and keep stability of fitting effects, significant items must be held, while the insignificant terms should be dropped. Therefore, step-wise regression method is adopted to simplify the PRSM. F–test can check each sub-item and evaluate whether it satisfies the reasonable conditions or not. If yes, the sub-item will be held; otherwise, it will be eliminated. The F–value for adding or deleting polynomials can be expressed as:

$$\mathrm{F} = \frac{RSS_{p'-1} - RSS_{p'}}{RSS_{p'}/(k - p' - 1)} \tag{6}$$

where $p'$ is the total number of polynomial functions in current approximation model, $k$ is the number of sampling points for construction the PRSM and the residual sum of square (RSS) is described as:

$$RSS_k = \sum_{i=1}^{k} (y_i - \hat{y}_i) \tag{7}$$

where $y_i$ are actual response values, $\hat{y}_i$ are approximate response values.

After eliminating the non-significant terms, the final fitting equation of displacement response are described as follow:

$$x_1 = -0.39 + 0.02v^2 + 1.57 \times 10^{-4} nmv^2 \tag{8}$$

$$x_2 = 0.64 + 0.52v + 0.03v^2 \tag{9}$$

$$\begin{aligned} x_3 = &-1.54 + 0.27m - 0.53n^2 - 0.004m^2 - 0.13nm + 0.04nv - 0.009\text{mv} - 0.05n^3 + 4.08 \times 10^{-5}m^3 \\ &+0.002nm^2 - 0.014m^2v + 1.87 \times 10^{-4}mv^2 + 5.17 \times 10^{-5}m^2v + 0.003nmv \\ &-4.47 \times 10^{-7}m^4 - 1.65 \times 10^{-4}n^2m^2 + 9.134 \times 10^{-4}n^3m + 8.38 \times 10^{-5}n^2v^2 \\ &+3.39 \times 10^{-4}n^3v + 1.42 \times 10^{-6}mv^3 - 2.21 \times 10^{-5}nmv^2 - 2.24 \times 10^{-5}nm^2v \\ &+8.109 \times 10^{-5}mn^2v \end{aligned} \tag{10}$$

$$\begin{aligned} x_4 = &-9.29 + 0.79m - 0.02m^2 + 0.001v^2 + 0.005nm + 3.85 \times 10^{-4}m^3 - 1.52 \times 10^{-4}nm^2 \\ &-4.87 \times 10^{-5}m^2v + 1.35 \times 10^{-5}m^2v + 2.46 \times 10^{-4}nmv - 2.05 \times 10^{-6}m^4 \\ &+7.88 \times 10^{-7}nm^3 + 5.14 \times 10^{-7}m^2v^2 - 2.87 \times 10^{-7}m^3v \end{aligned} \tag{11}$$

$$\begin{aligned} x_5 = &-62.72 + 6.64m - 0.23m^2 + 0.005mv + 0.003m^3 - 1.9 \times 10^{-5}m^4 + 9.76 \times 10^{-7}m^3v \\ &+8.48 \times 10^{-5}nmv^2 \end{aligned} \tag{12}$$

### 5.3. Error Estimation

To evaluate the precision of the response model, various indices such as Relative Error (*RE*), Root Mean Square Error (*RMSE*), and Coefficient of Determination ($R^2$) can be defined in Equations (13)–(15), respectively, as follows:

$$RE = 100\% \times \frac{(\hat{y}_i - y_i)}{y_i} \tag{13}$$

$$RMSE = \sqrt{\frac{\sum_{i=1}^{n}(\hat{y}_i - y_i)}{n}} \tag{14}$$

$$R^2 = 1 - \frac{SSE}{SST} = 1 - \frac{\sum_{i=1}^{n}(\hat{y}_i - y_i)}{\sum_{i=1}^{n}(\overline{y}_i - y_i)} \tag{15}$$

where $y_i$, $\hat{y}_i$ and $\overline{y}_i$ are the true values, predicted value, and mean values of the sample point, respectively; $n$ is the number of sample points.

Table 4 lists the evaluated results. Generally, for larger $R^2$ values, the surrogate model is more accurate. For a smaller *RE* and *RMSE*, the surrogate is better. It can be seen from Table 3 that the constructed surrogate models have a very high accuracy.

**Table 4.** Accuracies of the PRS model.

| Evaluation Type | RE (%) | RMSE (/) | $R^2$ (/) |
|---|---|---|---|
| $x_1$ | [−7.37, 8.40] | 0.598 | 0.9998437 |
| $x_2$ | [−2.24, 0.43] | 0.085 | 0.9999935 |
| $x_3$ | [−8.32, 9.23] | 0.051 | 0.9979243 |
| $x_4$ | [−5.98, 9.09] | 0.027 | 0.9991360 |
| $x_5$ | [−9.37, 5.79] | 0.406 | 0.9993867 |

To validate the accuracy of the established PRSMs, the relationship between the predicted and actual values of the displacement response are plotted in Figure 16. It can be found that the points of displacement response essentially distribute on both sides of the diagonal line. Thus, the regression model is well fitted with the observed values.

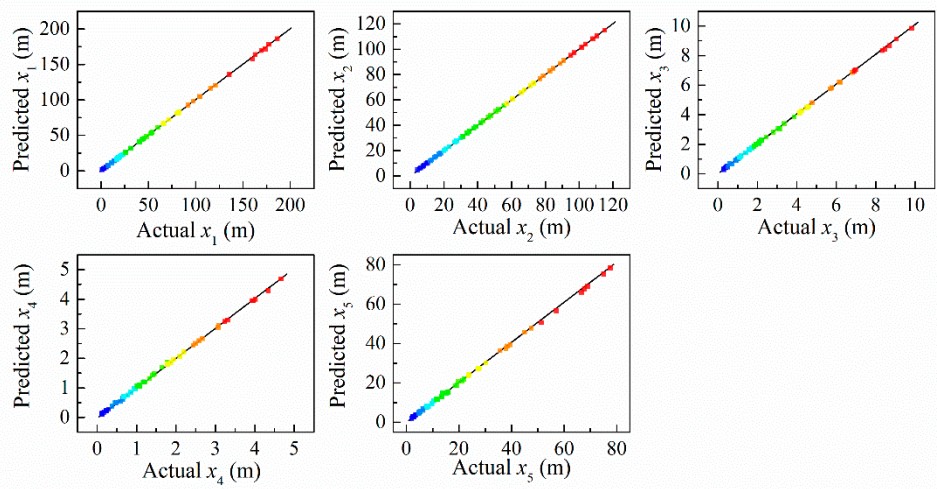

**Figure 16.** Scatter diagram of the displacement responses.

## 6. Conclusions

In this paper, the multi-body dynamics method is used to study the vehicle dynamic response at each stage of the train impact process under the influence of different parameters. The following conclusions are drawn:

1.  A 1D/3D co-simulation dynamics model of the train under various load conditions of driving, collision and braking has been established based on the MotionView dynamic simulation software. The accuracy of the numerical model is verified by comparing with a five-vehicle formations train-to-train crash test data.
2.  Based on the train dynamic model verified by test, this paper studied the effects of variable parameters on the train dynamic response at each stage of collision. The results show that the increase in the impact velocity has an increasing effect on the movement displacement of the vehicle in each process. However, increasing the vehicle mass and train formation has almost no effect on the running displacement of the braking process of the traction train. Through sensitivity analysis, we can find that the impact velocity has the greatest impact sensitivity for all test phase, followed by the train formation and vehicle mass;
3.  PRSM is used to construct the fitting relationship between the parameters and the responses, and step-wise regression method is adopted to simplify the PRSM. The fit-ting relationship can be applied to the design of the preliminary test plan for the train-to-train crash test, providing theoretical support for the test line length design, test equipment debugging and safety protection device installation, etc.
4.  According to the requirements of the standard EN15227 for train collision scenarios, the research completed in this paper only considers the first type of collision scenario. The future research will continue to conduct around the other two collision scenarios specified in the standard EN15227.

**Author Contributions:** Investigation, B.L.; Methodology, H.Z. and P.X.; Project administration, B.L.; Resources, P.X. and S.Y.; Software, H.Z.; Validation, W.G. and X.X.; Writing—original draft, H.Z.; Writing—review & editing, C.Y. All authors have read and agreed to the published version of the manuscript.

**Funding:** This work was supported by the National Natural Science Foundation of China [No. 51675537] and the Leading Talents in Science and Technology of Hunan Province in 2019 [No. 2019RS3018]. The financial support is gratefully acknowledged.

**Institutional Review Board Statement:** Not applicable.

**Informed Consent Statement:** Not applicable.

**Data Availability Statement:** Not applicable.

**Conflicts of Interest:** The authors declare no conflict of interest.

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
