# Peer review of "Full-Scale Train-to-Train Impact Test and Multi-Body Dynamic Simulation Analysis"

_machines, doi:10.3390/machines9110297_

Round 1

Reviewer 1 Report

The paper sounds to be good but not ready for publication. 

Authors are encouraged to add more technical content for the paper before further consideration. Technical content should include the mass of the train under each axle load, linear/nonlinear materials properties of the train main components (EI), some dimensions for the train cars and axle spacing, spring and damping coefficients, friction coeff. and etc. 

As for the finite element modeling; please indicate the name of the software used in the analysis, type of shell or solid elements, definition of the contact elements, number of elements as well as the DOFs, and how did the authors modeled the impact loadings. 

Graphs and tables are not enough to provide understanding, and there values have to be presented into the text with explanation. 

As the paper shows good research work, and it is all about how to present technical information. 

Author Response

Thanks for the reviewer's comments. The corresponding responses are attached below.

Reviewer 2 Report

The paper is about a simulation and validation of a complex machanical system. The study is considered as useful and after a major revision it would be an excellent article of the journal. The calculations are precise, however the presentation of the methodlogy and the results is poor:

Abstract:

In the abstract the main concrete results should be added in one or two short sentences.

Introduction:

The novelty of the study should be added.

Test:

In the introduction the authors structure the study as it is about a modelling, which is validated by actual test results. This means that logically the description of the simulation should be preceded the description of the test. This should be the methodological part of the paper. After that the results of the simulation (discussion/results) and at the end of the discussion the validation (physical test results) should be taken place. This means that the sections and subsections is highly recommended to be restructured. The reader can hardly follow what they are reading about at the moment: simulation or test results. It’s so confusing.

Fig.9. Tells nothing about the validation of the simulation. A description about the connection between the simulated and the test data would be useful.

Table 2: Error% in two rows with identical values. Why two rows?

3.3.2 is again about the comparison of the two types of data. The parallel handling of the two test results is confusing in this early discussion phase. Beyond this the graphs in Fig 10 are also confusing: many colored lines, one covered by another… Two separated figures would be clearer for both the velocity and the displacement.

Parametric analysis:

’…dynamic model verified by experiments…’: A model cannot be verified. The application or the results of it can be.

In 4.2 the effect of various masses is demonstrated. This means that in Fig 12 simulated data are demonstrated. However, the authors in the first sentence of the subsection write ’train crash test’. It is really misleading a piece of information. And again, boosts up the confusion in the reader.

4.3 Sensitivity analysis: the section tells almost nothing about the analysis. Please describe the parameters, the method, the meaning of the values, and the main data of the analysis. What n, m and v means here?

Section 5 is about a Doe-based RS determination. Describing Doe should be taken place earlier in a methodological section. However: DoE was not used in the simulation, there was no word earlier about 50 levels! Now, in this section the reader could think that there is another experiment within the original one. But this one is better because of the more detailed set-up (50 level). Moreover, there is no description or even a table about the combination of variables and formation of levels in the paper. Here in analyzing the response surfaces the physical test results are missing. Why the response surfaces if they are not connected to the real test results (even if partially)?

Methodology:

In the simulation model the simulation is poorly described. There are many input data that form the basics of the simulation. These are missing from the paper. The consequence of this is that the simulation cannot be reproduced by an independent expert, which means that the results are not well-founded! A detailed parametrization part is recommended to be added to the methodological section. Again, a clear methodology section with subsections (Simulation, Test, Doe) is highly recommended!

Conclusion:

The conclusion should contain the following and not more: clear, stand-alone sentences in the form of a list about the concrete results. The readers should understand these without reading the whole paper. The second is the summarization of the novelty and the applicability (academic and/or industrial) of the results. The third is the limitations of the study (this part can be placed to the intro): what things were neglected by the simulation. The fourth is the future research possibilities, extensions, etc.

Author Response

(The authors gave the same response as above.)

Reviewer 3 Report

Nice article, with excellent results. I recommend the authors to continue the research. The UNIZA Institute of Expert Research also deals with the issue of crash tests, but in automotive field. https://uzvv.uniza.sk/www2/index.php/sk/ 

Author Response

(The authors gave the same response as above.)

Round 2

Reviewer 2 Report

The authors improved the paper significantly. All the recommendations were considered and the necessary specifications or corrections were made. The paper requires a minor spell-check (English academic writing style) before the final submission.